# The Impact of Dental Care Programs on Individuals and Their Families: A Scoping Review

**DOI:** 10.3390/dj11020033

**Published:** 2023-01-30

**Authors:** Abdulrahman Ghoneim, Violet D’Souza, Arezoo Ebnahmady, Kamini Kaura Parbhakar, Helen He, Madeline Gerbig, Audrey Laporte, Rebecca Hancock Howard, Noha Gomaa, Carlos Quiñonez, Sonica Singhal

**Affiliations:** 1Faculty of Dentistry, University of Toronto, Toronto, ON M5G 1X3, Canada; 2Faculty of Dentistry, Dalhousie University, Halifax, NS B3H 4R2, Canada; 3Institute of Health Policy, Management and Evaluation, Dalla Lana School of Public Health, Toronto, ON M5T 3M7, Canada; 4Canadian Centre for Health Economics, Toronto, ON M5T 3M6, Canada; 5Department of Economics, University of Toronto, Toronto, ON M5S 3G7, Canada; 6Schulich School of Medicine and Dentistry, University of Western Ontario, London, ON N6A 3K7, Canada; 7Public Health Ontario, Toronto, ON M5G 1M1, Canada

**Keywords:** dental care programs, oral health outcomes, individual level, family outcomes, oral health, patient reported outcome measures, dental caries, dental care

## Abstract

Background: Despite significant global improvements in oral health, inequities persist. Targeted dental care programs are perceived as a viable approach to both improving oral health and to address inequities. However, the impacts of dental care programs on individual and family oral health outcomes remain unclear. Objectives: The purpose of this scoping review is to map the evidence on impacts of existing dental programs, specifically on individual and family level outcomes. Methods: We systematically searched four scientific databases, MEDLINE, EMBASE, CINAHL, and Sociological Abstracts for studies published in the English language between December 1999 and November 2021. Search terms were kept broad to capture a range of programs. Four reviewers (AG, VD, AE, and KKP) independently screened the abstracts and reviewed full-text articles and extracted the data. Cohen’s kappa inter-rater reliability score was 0.875, indicating excellent agreement between the reviewers. Data were summarized according to the PRISMA statement. Results: The search yielded 65,887 studies, of which 76 were included in the data synthesis. All but one study assessed various individual-level outcomes (n = 75) and only five investigated family outcomes. The most common program interventions are diagnostic and preventive (n = 35, 46%) care, targeted children (n = 42, 55%), and delivered in school-based settings (n = 28, 37%). The majority of studies (n = 43, 57%) reported a significant improvement in one or more of their reported outcomes; the most assessed outcome was change in dental decay (n = 35). Conclusions: Dental care programs demonstrated effectiveness in addressing individual oral health outcomes. However, evidence to show the impact on family-related outcomes remains limited and requires attention in future research.

## 1. Introduction

Oral diseases are one of the most common preventable chronic diseases. More than three billion people worldwide suffer from dental caries, severe periodontal disease, tooth loss, and oral cancer [1]. However, with the exception of periodontal diseases [1], other oral health problems such as dental caries and tooth loss have significantly declined over the last few decades [2]. This trend is observed across all age groups in a number of the world’s middle- and high-income countries such as Australia, Brazil, Canada, Japan, Nordic and Western Europe, and the United States (US) [3,4].

Despite such improvements, the burden of oral disease continues to concentrate in individuals who are socially and/or economically disadvantaged, including those living in poverty, with developmental or acquired disabilities (e.g., frail and functionally dependent older adults [5,6]), and in rural and remote regions (e.g., indigenous populations). This results in both oral health-related inequities and inequalities. The societal and economic impacts of untreated dental diseases can be significant. In recent estimates, the global direct and indirect costs of oral diseases amount to USD 298 billion and USD 144 billion per annum, respectively [7]. In addition to the economic losses associated with oral diseases, other intangible impacts such as missed school days, poor academic performance, hindered familial and social interactions, inability to work, and lost productivity are some of the documented societal consequences of poor oral health [8,9,10,11,12,13,14,15].

Even in some high-income nations with universal health care, access to essential dental services has been proven challenging, such as in Australia, Canada, and Italy. According to a comparative study conducted by Allin et al. [16], of the eight countries examined, the proportion of public funds of total dental expenditure ranged from 5.8% to 40.2%, with the lowest being in Australia, Italy, and Canada at 18.0%, 8.6%, and 5.8%, respectively. The highly privatized nature of these dental care systems is often to blame for the well-documented oral health inequities and the lack of access to essential dental services for those unable to pay [17,18].

Dental care programs can potentially increase access to services, improve utilization of dental care, and likely have positive impacts on oral health-related outcomes at the individual and family levels. In a study conducted by Zivkovic and colleagues, data from a self-administered survey in Ontario, Canada’s most populous province, were analyzed to understand the impact of having dental insurance on self-reported oral health and dentist visiting behaviors [19]. The authors found that those with dental insurance were more likely to report very good or excellent oral health, more likely to utilize routine dental services, and less likely to visit the dentist for emergency services [19]. Manning and colleagues estimated the effects of dental treatment (except orthodontics) through cost-sharing dental insurance plans (with varying proportions of out-of-pocket payment) on oral health outcomes of low-income families in six cities across the US in the 1970s [20]. The exit oral examination showed people in the free dental plan to have better oral health outcomes than those in the other plans. The most significant differences were in adolescents, who had significantly fewer decayed teeth and a lower periodontal index score [21]. Other studies have also shown that receiving preventive or curative dental interventions can prevent tooth decay and reduce the number of untreated carious lesions and emergency department (ED) visits for non-traumatic dental problems, as well as improve quality of life (QoL) and social functioning [8,9,22,23,24].

Similarly, community dental clinics (CDC) that provide dental services to the socially disadvantaged have also been identified as a viable targeted oral health intervention [25]. These avenues aim to address oral health inequities by enabling access to dental services for low-income individuals. Such programs present a potential opportunity for integrating oral health in primary healthcare settings to reduce the incidence of oral health and related general health problems. However, Wallace and colleagues argue that, despite its conceptual soundness, the lack of evidence to support the effectiveness of such avenues and the disinterest of governments to finance them limits the large-scale adaptation of CDC programs [25]. Even in jurisdictions where the political climate may be in favor of implementing a dental care program, several questions remain unanswered, thus hindering the progress of such initiatives. Frequently asked questions include what services ought to be included in these programs, what evidence exists in terms of effectiveness, and whom these programs should target. Finding answers to these and other related questions can help inform future oral health programs and tackle oral health-related inequity.

The relationship between access to dental care and enhanced oral health seems intuitive; however, the effects of dental care programs at the individual and family level remain largely unexplored. Implementing dental care programs requires political will and support from policy makers, which may be driven by evidence on the value of implementing such programs for individuals, families, society, and the economy. To advance the notion of equitable access to dental care, evidence supporting the benefits of dental programs must be demonstrated. A scoping review was planned to explore the effectiveness of dental care programs in reducing oral disease in high-income countries, specifically to answer the following questions: (i) what is the scope of the existing literature (i.e., the type of services provided, target populations, implementation settings) assessing dental care programs, (ii) what measures have been used to assess the impact of dental care programs on individuals and families, and (iii) how effective are these programs in improving oral health and related outcomes?

## 2. Methods

### 2.1. Conceptual Framework

The conceptual framework used to inform this scoping review is the one described by Graham [26] in their analysis of the interventions utilized to tackle health inequalities in England. One of the examined approaches was the targeted interventions that focus on the socially disadvantaged. It is believed that, by targeting the worst off in societies, the health gap between them and the more well off will narrow and produce more equitable health and oral health outcomes. For many years, targeted interventions have been an important driver of oral health policies leading to the planning and implementation of various oral health programs.

### 2.2. Design and Study Search

Scoping reviews are a relatively new approach to synthesize evidence [27]. Unlike systematic reviews, they examine the extent and nature of heterogenous bodies of evidence to identify gaps in the literature and guide future research. Given their relative novelty, current guidelines for reporting scoping reviews do not yet exist. Nonetheless, guiding documents and good reporting practice published by the Joanna Briggs Institute (JBI) [28] and Tricco et al. [29] are essential to ensure consistent and high quality scoping reviews. 

This scoping review with a narrative synthesis was conducted in accordance with the PRISMA statement for transparent reporting of systematic reviews and meta-analysis [30] and the *JBI Reviewers Manual* [28]. With the help of two librarians (HH and MG) at the University of Toronto library, search strategies were developed and run in four online databases, MEDLINE, EMBASE, CINAHL, and Sociological Abstracts, on November 18, 2021. Articles from December 1999 to November 2021 were included. The full search strategy is outlined in Appendix A. The inclusion of studies published after December 2019 allowed us to assess the effect of dental programs during the COVID-19 pandemic. The protocol was drafted and reviewed by the research team and can be accessed by contacting the corresponding author.

### 2.3. Inclusion Criteria and Screening Process

We only included those dental care plans/programs that provided dental care interventions of some form, whether privately (e.g., enrolling in a dental program based on certain eligibility criteria) or through public funding (e.g., expansion of Medicare and Medicaid to include dental services) with the aim to improve oral health outcomes. We only included studies conducted in high-income countries according to the World Bank data based on their respective gross national income (GNI) [31]. For systematic reviews and meta-analyses, which included studies from all income countries, we specifically included only those from high-income countries; however, if country information was missing, we included it to avoid missing potential evidence. Other inclusion criteria were studies that assessed outcomes at the individual and/or family level and that were available in English and published after 1999. We excluded studies that were purely qualitative in nature, assessed isolated dental interventions without being part of any project or a program, focused on population level interventions (e.g., water fluoridation and tobacco cessation programs), grey literature, conference abstracts, and preprints. Furthermore, to accommodate the nature of our research question, we limited our inclusion of oral health education (OHE) programs to those geared toward patient populations (i.e., excluded studies where educational programs were targeting healthcare providers such as physicians, nurses, etc.). No other restrictions were placed on sample size, targeted populations, age, study design, or the types of services included in the program.

### 2.4. Selection and Data Extraction

All identified articles were imported into Covidence, an online software used for managing and streamlining systematic reviews. De-duplication was conducted through the systematic process described by Bramer et al. [32]. Four independent reviewers (AG, AE, VD, and KKP) screened the titles and abstracts for their relevance. To ensure consistency across all reviewers, Cohen’s kappa inter-rater reliability score was calculated on a random sample of twenty studies retrieved from the study pool; this yielded a score of 0.875, indicating excellent agreement. Any disagreements were resolved through a discussion until consensus was reached. Full-text articles were retrieved from the University of Toronto online library, Google Scholar, and Research Gate. Full-text review was performed and only studies that fulfilled the eligibility criteria underwent data syntheses. If a study was part of an included review (systematic review, etc.), it was excluded to prevent redundancy. Similarly, if one study reported the same outcome at multiple time points through a number of publications, the most recent report was included in the synthesis. Finally, quality appraisals were not conducted for the studies included. This was carried out to align with the scope of this review to map the breadth of evidence but not the depth. Furthermore, we believe that imposing quality measures can omit important studies that can help inform future programs. 

## 3. Results

In total, 93,603 records were identified through the database search and imported into Covidence. De-duplication resulted in 65,887 unique records for screening, of which 76 studies were included in this analysis (Figure 1. PRISMA flowchart). Of the included studies, fifty-six studies (75%) were published in 2010 or later. Thirty-five studies (46%) examined the outcomes of diagnostic and preventive programs, forty-two studies (55%) were conducted among children, and twenty-eight studies (37%) were conducted in school-based settings. The majority of studies were conducted in European (n = 35, 46%) and North American (n = 16, 21%) countries. The most common study designs were experimental (n = 49, 46%) and prospective cohort studies (n = 10, 13%). Table 1 outlines the descriptive characteristics of the studies included in this review.

### Data Synthesis

The included studies are categorized based on the type of program area: oral health education; diagnostic and prevention; dental care intervention (Appendix A provide a summary of these). For each program area, the results are presented by (1) type of services provided; (2) the setting, target population(s), and personnel involved; (3) outcomes measures and the reported impacts of the program. A detailed description of the breakdown is presented in Table 2. Figure 2 depicts the visual breakdown of the described categories. 

A. Oral Health Education (OHE) Programs 

A.1 Types of Services Provided

Of the included studies, 24 original studies [34,35,36,37,38,39,40,41,42,43,44,45,46,47,48,49,50,51,52,53,54,55,56,57] and five systematic reviews [58,59,60,61,62] evaluated the effectiveness of OHE programs. The type of OHE program implemented was further categorized according to the teaching domains introduced by Bloom in 1956 [33], namely cognitive, psychomotor, and affective. The cognitive domain involves the acquisition of knowledge and the development of intellectual skills such as the recognition of specific facts, procedural patterns, and the ability to recall or retrieve previously learned information. The psychomotor domain focuses on the physical movement, coordination, and use of motor-skills. Finally, the affective domain of learning includes how participants deal with things emotionally, such as enthusiasm and motivation. The cognitive domain (n = 9) [35,37,38,39,41,48,49,51,55] alone and the combination of cognitive and psychomotor (n = 7) [40,43,45,47,50,54,56] were frequently utilized for conveying the oral health education message. Table 3 provides a detailed breakdown of the domains of learning applied in the studies. 

A.2 Setting, Target Populations, and Personnel Involved

The most common settings for delivering the OHE program were community centers (n = 9), followed by schools (n = 8). Most programs were geared towards children and adolescents (n = 18) and were either delivered solely by dental personnel (i.e., dentists and dental hygienists) (n = 12) or with the help of healthcare or non-healthcare personnel (n = 6).

A.3 Outcome Measures

A.3.1 Dental Caries

Ten studies explored the effectiveness of OHE programs in reducing dental caries, with the majority (n = 6) reporting improved outcomes [34,37,40,46,55,62]. However, one study [53] reported no impact and one study [42] and two systematic reviews observed inconclusive findings [59,61].

A.3.2 Gingival Health and Oral Hygiene

Thirteen studies reported the impacts of OHE programs on either the individuals’ gingival health, their oral hygiene, or both. Of these, nine studies [34,41,43,46,47,49,57,60,61] reported a significant improvement in one or both outcomes, three studies reported inconclusive results [35,45,59], and one reported no significant changes [38].

A.3.3 Oral Health-Related Quality of Life (OHRQoL), Oral Health Knowledge, Attitudes, and Behaviors

One original study [44] and one systematic review [58] assessed the impact of OHE programs on OHRQoL measures and both demonstrated positive impacts. Two systematic reviews and 12 studies examined the impacts of OHE programs on participants’ oral health knowledge, attitudes, and behaviors [34,38,39,44,46,47,48,50,51,52,53,54,59,61]. Of these, 11 reported positive outcomes and only 3 studies reported inconclusive or insignificant improvements [38,39,53].

A.3.4 Miscellaneous Clinical Outcomes

Two studies and a systematic review examined other clinical outcomes such as salivary flow, oral function, and halitosis [36,41,58]. All the programs resulted in increased oral function, salivary flow, and reduction of halitosis.

A.3.5 Family Level Outcomes

One systematic review and one original study assessed the impact of OHE programs on oral health knowledge, attitude, and behaviors [56,61] of parents; both demonstrated a significant positive impact.

B. Diagnostic and Preventive Programs

B.1 Types of Services Provided

Thirty-five studies assessed the impact of a wide array of primary and secondary preventive dental services [63,64,65,66,67,68,69,70,71,72,73,74,75,76,77,78,79,80,81,82,83,84,85,86,87,88,89,90,91,92,93,94,95,96,97]. These preventive services were either delivered singularly or in combination with others. The most common preventive agent used is fluoride varnish. Eighteen studies provided fluoride varnish solely or in combination with other preventive modalities such as chlorohexidine gel, sealants, and non-invasive therapies [63,64,65,67,68,69,70,71,72,75,76,77,85,87,89,90,95,96]. Other common forms of preventive programs used were fluoride gel [74,78,94], fluoride mouth rinse [82,84,95], fluoride tablets [73,81,97], and professional debridement [79,91,93,94,96].

B.2 Setting, Targeted Populations, and Personnel Involved

The majority of preventive programs included in this review were school based (n = 18) programs and targeted children (n = 21). Other settings included community/public health settings (n = 8) [63,65,69,73,81,83,87,97], medical settings (n = 5) [66,79,90,93,95], and dental settings (n = 3). Most of the programs were delivered by dental personnel (n = 26), four programs were solely implemented by healthcare and non-healthcare personnel [65,66,78,80], and six were unclear on who provided the intervention [67,71,74,75,82,92].

B.3 Outcome Measures

B.3.1 Dental Caries

The most common outcome assessed in the studies included was dental caries. Of the 31 studies that reported caries-related outcomes, 28 reported a positive impact, 2 reported no significant changes [70,95], and 1 reported mixed results [72].

B.3.2 Gingival Health and Oral Hygiene

Of the included studies, seven studies assessed gingival health, periodontal health, and oral hygiene practices. Six studies reported positive outcomes in one or more of the reported indices [78,80,83,91,93,95] and one found no significant changes attributed to the program [73].

B.3.3 OHRQoL, Oral Health Knowledge, Attitudes, and Behaviors

Seven studies examined the impact of diagnostic and preventive programs on participants’ oral health knowledge, attitudes, and oral hygiene behaviors. All but two reported significantly improved outcomes [73,89]. One study examined the impact of the program on the participant’s OHRQoL using the early childhood oral health impact scale (ECOHIS). The study found that children enrolled in the early head start (EHS) program have shown significant improvements in OHRQoL of children (i.e., reduction in pain, improvements in eating and other functions) compared with their counterparts who have not been enrolled in the program [92]. 

B.3.4 Miscellaneous Clinical Outcomes

Other outcomes of interest were dental fluorosis [97], salivary secretion, and quality of life (QoL) [95]. Eckersten et al. reported that children who participated in the oral health program including fluoride tablet supplements from the age of 2 years did not report significantly different levels of dental fluorosis on their permanent teeth compared with those who did not enroll in the program [97]. Finally, the study conducted by Lee et al. found that providing OHE in conjunction with fluoride varnish and fluoride mouth rinse improved the salivary flow and the quality of life of patients undergoing radiotherapy for head and neck cancer [95].

B.3.5 Family Level Outcomes

Two studies examined the impact of preventive programs on family-level outcomes. Schroth et al. [87] found that the involvement of the parents in the preventive programs tailored for their children resulted in significant improvements in their oral health knowledge and attitudes and mixed results were reported regarding their oral health behaviors [87]. Burgette and colleagues also found that the children’s enrolment in the early head start program resulted in a significant improvement in parents’ distress and family function subscales of the ECOHIS [92].

C. Dental Care Interventional Programs

C.1 Types of Services Provided

As aforementioned, dental care interventional programs are those providing the most comprehensive range of services with a focus on tertiary prevention or curative care. These included, but were not limited to, restorations, extractions, and dental prosthetic treatment. Of all the studies included in the review, only 12 investigated the impact of providing therapeutic and curative dental services [98,99,100,101,102,103,104,105,106,107,108,109] on individual and family level outcomes. Seven programs outlined the services provided, which were mostly scaling, simple restorations, and extractions. Table 4 outlines the full list of services provided [101,103,104,105,106,108,109].

C.2 Setting, Target Populations, and Personnel Involved

The two most common settings in which the included studies were implemented were school (n = 6) and community based (n = 3). Nine programs were geared towards adults and elders, more specifically: welfare recipients [103,104], pregnant women [105,106], and institutionalized elders [102,108]. Nine programs were delivered solely by dental personnel [99,100,101,103,104,105,107,108,109], two with the aid of healthcare personnel (i.e., midwives and nurses) [102,106], and one did not specify the type of personnel involved [98].

C.3 Outcome Measures

C.3.1 Dental Caries

The studies investigating the impact of these dental care interventions used a wide range of clinical and self-reported outcomes. All six programs assessing dental decay as an outcome [99,102,105,106,107,108] reported a significant reduction in caries rates. 

C.3.2 Gingival Health and Oral Hygiene

Two studies investigated the impact of interventional dental care on oral hygiene and gingival and periodontal health and reported significant improvements [106,108]. 

C.3.3 OHRQoL, Oral Health Knowledge, Attitudes, and Behaviors

Six studies utilized various instruments to capture the changes in OHRQoL among children, adults, and elders. Hyde and colleagues [104] and Ortuno Borroto et al. [109] employed the oral health impact profile (OHIP) instrument in their study. Walker et al. [101] and Rong et al. [108] captured the changes in OHRQoL due to dental treatment of their elder population using the general oral health assessment index (GOHAI). Additionally, in two studies, Alsumiat et al. [98,99] used individual questions to evaluate the impact of dental care treatment programs on the OHRQoL of children using questions that were not part of validated instruments. 

Changes in oral health knowledge, attitude, and behaviors were assessed in three studies. Two studies reported a significant improvement in self-reported oral health, knowledge about oral health, an increased likelihood of visiting the dentist, and improved patient satisfaction with treatments [103,106]. On the contrary, Alsumiat et al. [98] reported no significant differences in the oral hygiene knowledge and practices between those who attended and those who did not attend the dental program. 

C.3.4 Miscellaneous Clinical Outcomes

Only two studies examined the impact of interventional dental care on non-conventional clinical measures. Both reported positive outcomes, including significant improvements in clinical oral disorders [100] and oral health stability [102]. 

C.3.5 Family Level Outcomes

One study investigated the impact of a dental care intervention at the family-level outcomes [99]. Alsumiat et al. [99] investigated the impact of providing OHE on mothers in conjunction with providing dental treatment to their children. They reported no impact on mothers’ oral health knowledge, attitude, practices, or OHRQoL. 

## 4. Discussion

This scoping review has synthesized the published literature investigating the impact of dental care programs on oral health outcomes at the individual and family level in high-income countries. The study also identifies knowledge gaps in the literature around interventions and population groups. To our knowledge, this is the first study to summarize the evidence describing dental care programs regardless of their scope. Even in high-income countries with universal healthcare systems, oral health inequities persist and socially disadvantaged populations carry a greater burden of oral diseases. In those settings, many turn towards targeted dental care programs as a viable option to narrow those differences [17,110]. With more policymakers turning toward evidence to support decisions, this addition to the literature will potentially aid in evidence-informed decisions on the nature and scope of future dental programs. 

This review makes several important observations. First, despite not being the most popular form of dental programs identified, OHE remains an extremely popular form of dental program in the last two decades despite the uncertainty about its long-term effectiveness [111,112]. It is well-known that health is strongly tied to several socioeconomic and individual factors, deeming OHE programs hard to evaluate in isolation of these attributes. Nonetheless, OHE being part of “developing personal skills”, one of the Ottawa Charter principles [113], and, given the relatively low cost associated with these programs, they appeal to many policy decision makers. Second, the significantly smaller proportion of programs targeted toward adults and elders suggests a clear deficiency in the literature and in the interventions that policymakers and program planners are willing to support and implement. This deficiency interferes in understanding the effectiveness of dental programs delivered to adults, especially to those who face social and economic challenges. 

Furthermore, our review highlights the relatively small number of studies assessing the effectiveness of curative dental programs. The broad scope and consequently high price tag perhaps explain the reason for their unpopularity. That said, given the growing interest in the impact of interventional dental care on the subjective psychosocial well-being of patients, it was not surprising that we were able to identify six studies that utilized various instruments to capture the changes in OHRQoL among children, adults, and elders. Finally, despite the plausibility of an association between improved oral health and family outcomes, the paucity of studies assessing the family impact of receiving dental treatment was surprisingly limited. Future research should attempt to address this knowledge gap. Other research questions to be considered are the impacts of dental programs on self-confidence, family relationships, and employability, which are often, but not always, captured within the identified OHRQoL measures. 

In general, the available evidence suggests that all types of programs demonstrate a positive impact on clinical and self-reported outcomes such as dental decay, periodontal diseases, oral health behaviors and knowledge, and, to a lesser extent, OHRQoL. However, it is important to note that, due to the ubiquitous heterogeneity in the timeframes of outcome evaluation (i.e., program length and follow-up intervals) and that one third of identified studies (n = 23) has follow-up intervals of 1 year or less, the longevity of effectiveness must be interpreted with caution. Nonetheless, aside from the apparent bias for assessing the impacts of conducting OHE programs on individual level outcomes, the design, outcomes, and evaluation timepoints suggest the methodological validity of the included studies and, therefore, the findings. 

Successful outcomes were also consistently high across all interventions delivered, regardless of the personnel. However, given the variation in medical and dental personnels’ scope of practice across different jurisdictions, replicating successful programs locally can present challenges. Furthermore, a considerable number of studies lacked clarity around some of the crucial details, particularly regarding the types and the personnel involved in the provision of services. It was interesting to note the comparable success rates between programs solely delivered by dental and non-dental personnel (54% vs. 45%, respectively). This has significant professional and cost implications that supports the integration and delivery of certain dental care programs in various public health and community settings by non-dental personnel. However, this might be challenged by professional regulations and differences in scope of practice across jurisdictions. 

A number of limitations should be taken into consideration. First, this review does not speak to the quality of the evidence but rather the nature and breadth of the studies that investigated the impact of dental programs on individuals and families. As the relevant literature was already scarce, we did not want to limit it further by excluding studies based on quality appraisal. Second, the studies were conducted in various jurisdictions, which likely have different political, legislative, and policy bases regarding how the programs are financed, administered, and delivered, and who are targeted for interventions; additionally, the socio-cultural environment would have affected the outcomes differently. Thus, the feasibility of a program and its outcomes in one population or jurisdiction may not necessarily be applicable in others. It is also important to note that, apart from the clinical effectiveness of dental programs, cost remains a relevant criterion for policy decision makers when endorsing any program. However, given the scope of this review, we did not investigate the cost-effectiveness of the identified dental programs. Nevertheless, it is of utmost importance for policy decision makers to be informed about the financial feasibility of recommended programs. Other relevant studies may not have been included in our review for various reasons, such as being incorrectly tagged, missed by our search strategy, or simply because it was published outside of our search window. Finally, despite the global impact that the COVID-19 pandemic had on dental care, we did not identify any studies that were relevant to the pandemic.

## 5. Conclusions

In summary, it appears that the majority of the dental programs included in this review reported success in improving oral health and related outcomes despite the reported differences in scope, type of intervention, personnel delivering the program, and the jurisdiction in which the program was implemented. Diagnostic and preventive programs were the most common programs implemented and children were the most targeted group of recipients. The results of this scoping review can help inform future dental programs, while underscoring knowledge gaps around several aspects of dental programs, specifically with regard to their benefit, long-term effectiveness, and impact on family-level outcomes.

## Figures and Tables

**Figure 1 dentistry-11-00033-f001:**
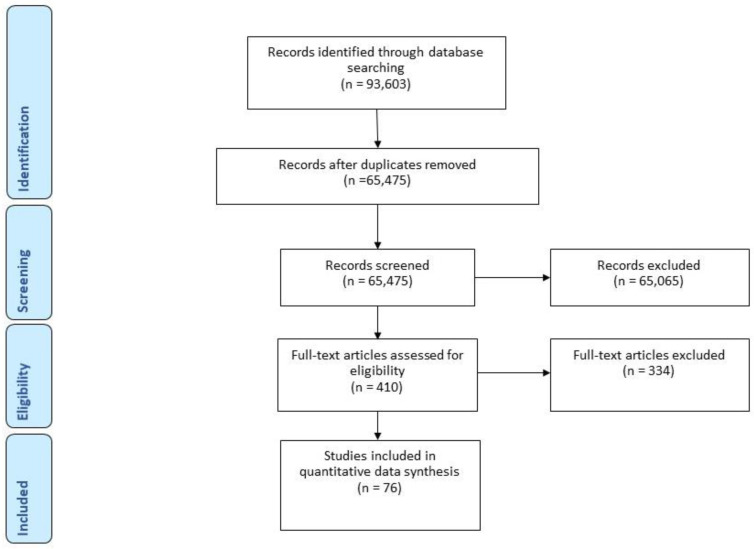
PRISMA flow chart.

**Figure 2 dentistry-11-00033-f002:**
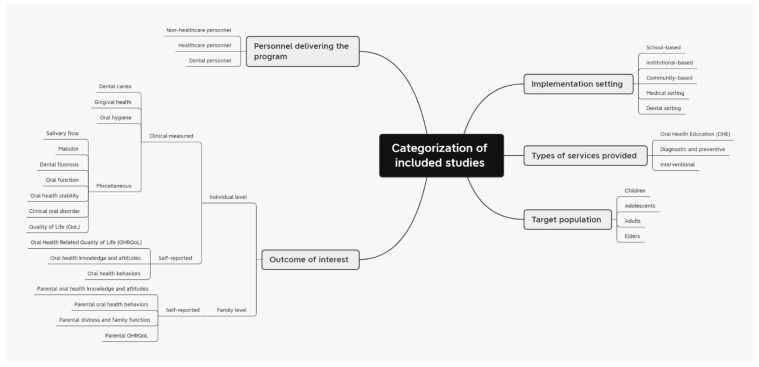
Categorization of the included studies.

**Table 1 dentistry-11-00033-t001:** Description of the studies included.

Characteristics of the Studies	n	%
Year of publication		
2000–2005	8	11
2006–2010	12	16
2011–2015	18	24
2016–2021	38	50
Region of study		
North America	16	21
South America	2	2
Europe	35	46
Middle East	4	5
Asia	8	11
Australia and Oceania	6	8
Not applicable	5	7
Type of intervention		
Oral health education (OHE)	29	38
Diagnostic and preventive	35	46
Interventional	12	16
Program setting *		
School-based setting	28	37
Long-term care and institutional settings	8	11
Community setting/public health setting	20	26
Medical setting	11	14
Dental setting	7	9
Reviews	5	6
Target population *		
Children	42	55
Adolescents	11	14
Adults	13	17
Elders	14	18
Not specified/unclear	4	5
Personnel delivering the program *		
Non-healthcare personnel	10	13
Healthcare personnel	13	17
Dental personnel	47	62
Not specified/unclear	11	14
Type of outcome assessed		
Clinical outcome(s) only	46	61
Behavioral/self-reported outcome(s) only	14	18
Both clinical and behavioral/self-reported outcomes	16	21
Study design		
Experimental study	47	62
Cross-sectional study	6	8
Longitudinal/prospective cohort	9	12
Retrospective cohort	5	7
Secondary analysis of data	1	1
Ecological study	1	1
Systematic review	5	7
Not specified	2	2
Outcomes measures *		
Individual level outcomes		
Caries	45	59
Gingival health	16	21
Oral hygiene	17	22
Oral health related quality of life (OHRQoL)	9	12
Oral health knowledge and attitude	15	20
Oral health behaviors	19	25
Miscellaneous	7	9
Family level outcomes		
Parental oral health knowledge and attitude	4	5
Parental oral health behaviors	4	5
Parental stress and family function	1	1
Parental OHRQoL	1	1

* Percentages might add to more than 100% as they can be included in the same study.

**Table 2 dentistry-11-00033-t002:** Description of the breakdown categories.

**A.** Implementation Setting	
School-based programs	Organized by schools and implemented either within schools or other educational premises, which include pre-school and public daycare centers.
Long-term care and institutional programs	Implemented in nursing homes, long-term care facilities, or residential homes for elders or adults with physical or mental challenges (e.g., cerebral palsy, neurodevelopmental disabilities, etc.).
Community-based programs	Implemented in public health/community centers. These include, but are not limited to, mother and child health centers, child health centers, and public health units.
Medical setting programs	Delivered in a medical context outside conventional dental settings, such as programs provided in hospitals, medical clinics, well-baby clinics, or other non-dental healthcare settings.
Dental setting programs	Programs implemented in conventional dental settings (i.e., private dental clinics) or dental schools.
**B.** Type of Services Provided *	
OHE programs	Programs that provide oral health education about preventing and managing various oral health diseases. It also includes oral hygiene instructions on the proper brushing techniques, the use of fluoridated toothpaste, and flossing. These were delivered in the form of lectures, motivational interviews, pamphlets, amongst other methods of oral health promotion. The programs utilize one or more of the three learning domains (cognitive, psychomotor, and affective) introduced by Bloom in 1956 [33].
Diagnostic and preventive programs	Includes those programs providing primary and secondary prevention services that aim to prevent or limit the development of oral diseases such as oral screenings and referrals, fluoride applications, pit and fissure sealants, and non-invasive therapies (e.g., atraumatic restorative treatment and interim stabilization therapy).
Interventional programs	Includes those programs that provide curative dental services such as restorations, extractions, root canal treatments, and prosthodontic care.
**C.** Targeted Population	
Children	Less than 12 years old.
Adolescents	Between 12 and 19 years old.
Adults	Between 20 and 64 years old.
Elders	Sixty-five years old and above.
**D.** Personnel Responsible for Delivering the Program	
Non-healthcare personnel	Individuals outside the healthcare sector such as teachers, caregivers, social workers, and trained peers.
Healthcare personnel	Individuals who work in the healthcare sector but do not have formal training in dentistry, such as nurses, physicians, and midwives.
Dental personnel	Individuals who have completed formal training in dentistry or dentistry-related fields such as dentists, dental hygienists, dental assistants, dental therapists, and dental nurses.
**E.** Outcome of Interest	
E.1 Individual outcomes	Outcomes that are specific to the individual receiving the intervention. Examples include changes in the oral health status, knowledge, and attitudes towards dental care behaviors.
E.1.1 Clinical outcomes	Assessed by calibrated examiners via clinical measures such as caries indices, periodontal measures, etc.
**1.** Dental caries	e.g., dmft/DMFT/, caries increment, percent increase in caries, demineralization, etc.
**2.** Gingival health	e.g., gingival and periodontal indices such as community periodontal index for treatment needs (CPITN), gingival index (GI), periodontal index (PI), bleeding on probing (BoP), clinical attachment loss (CAL), etc.
**3.** Oral hygiene	Assessed using indices for measuring plaque, calculus, debris, denture plaque, mucosal plaque score (MPS), etc.
**4.** Miscellaneous clinical outcomes **4.1** Salivary flow **4.2** Dental fluorosis **4.3** Oral health stability **4.4** Clinical oral disorder **4.5** Quality of life (QoL)	Other less commonly used clinical outcomes that are captured in the review include: – Measures of changes in the levels of salivary flow. – As measured by Dean’s index, Thylstrup and Fejerskov (TF) index, and other dental fluorosis indices. – Absence of emerging dental problems such as dental decay. – An amalgamated index that describes the oral health status by assessing caries and periodontal diseases and other aspects of the dentition. – The subjective perception of physical, emotional, social, and cognitive aspects that impact the quality of life.
E.1.2 Self-reported outcomes	Collected through chairside interviews or questionnaires without a clinical assessment.
**1.** Oral health related quality of life (OHRQoL)	Indices or measures that assess the impact of dental problems on the quality of life, such as the oral health impact profile (OHIP), oral impact on daily performances (OIDP), and geriatric oral health assessment index (GOHAI).
**2.** Oral health knowledge and attitude	All aspects of knowledge around the process of dental decay, periodontal disease, the importance of dental visits, fluoride agents, etc.
**3.** Oral health behaviors	Oral hygiene measures (frequency of brushing, flossing etc.), dietary habits including the consumption of sugary food and beverages, and the consumption frequency reported by the individuals.
E.2 Family level outcomes	Outcomes that are specific to the parents and the family of the recipients of the intervention.
**1.** Parental oral health knowledge and attitude	Measures of changes in the knowledge and perceptions about different dental disease processes.
**2.** Parental oral health behaviors	Measures change in parental oral health behaviors (e.g., brushing, flossing, etc.)
**3.** Parental distress and family function	Assesses change in parental distress levels attributed to changes in their dependents’ oral health status and wellbeing.
**4.** Parental OHRQoL	Measures change in parental quality of life influenced by their dependents’ oral health status and wellbeing.

* In cases where combinations of the services above were provided, the study was categorized based on the more comprehensive services. For example, if a program provided OHE in addition to fluoride varnish, it was categorized as a preventive program. Similarly, if a program offered a fluoride therapy along with any form of dental treatment such as restorations or extractions, it was categorized as an interventional program. This assumed that basic services are commonly included by default in programs that provide more comprehensive services.

**Table 3 dentistry-11-00033-t003:** The categorization of OHE programs based on their learning domain.

The Domain Addressed	n	%
Cognitive only	9	31
Psychomotor only	1	3
Cognitive, psychomotor	7	24
Cognitive, affective	4	14
Psychomotor, affective	2	7
Cognitive, psychomotor, affective	1	3
N/A	5	17

**Table 4 dentistry-11-00033-t004:** The breakdown of services provided with the interventional programs.

Author, Year	Services Provided
Alsumiat et al., 2015	Not specified
Alsumiat et al., 2019	Not specified
George et al., 2018	– Dental restorations – Denture assessments
Gomez et al., 2001	– Restorations – Extractions – Other preventive and diagnostics
Hyde et al., 2005	– Examination and X-rays – Prophylaxis – Periodontal scaling and root planing – Amalgam and composite restorations – Stainless steel crowns for molar teeth – Porcelain crowns for anterior teeth – Bridges for anterior teeth – Bleaching trays and nightguards – Dentures and all-acrylic partials – Endodontic root canals – Oral surgery – Nitrous oxide and sedation
Hyde et al., 2006	
Janssens et al., 2018	Not specified
Larsen et al., 2016	Not specified
Ortuno Borroto et al., 2021	Prevention, conventional periodontal treatments, dental fillings, root canal treatments, dental extractions, and removable dentures
Rong et al., 2009	– Extractions – Scaling – Fillings
Walker et al., 2007	– Simple and complex restorations – Extractions – Relining or constructing new dentures – Periodontal disease and prophylaxis services
Wyatt et al., 2009	Not specified

## Data Availability

All data is available in the article and Appendix A.

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
