# Peer review of "The Impact of Dental Care Programs on Individuals and Their Families: A Scoping Review"

_dentistry, 2023, doi:10.3390/dj11020033_

Round 1

Reviewer 1 Report

Dear authors, thank you for the interesting study. It is conducted at a high methodological level; it concerns the topical issue of the prevention of dental diseases at the individual and family level.

It seems to me that the manuscript will be of interest to readers and I can recommend accepting it in the present form.

Author Response

Thank you for your comments. We appreciate your feedback.

Reviewer 2 Report

Very relevant manuscript with relevant health policy issues. Stakeholders and policy makers need to assess the validity and reliability of scoping reviews like this one.

Authors must clearly distinguish between a systematic review (therefore using prisma usual guideline) from a scoping review (using PRISMA statement). Nevertheless, and because guidance for scoping review protocols does not yet exist, we acknowledge the effort of authors to better structure the review [Tricco, A. C., Lillie, E., Zarin, W., O'Brien, K. K., Colquhoun, H., Levac, D., ... & Straus, S. E. (2018). PRISMA extension for scoping reviews (PRISMA-ScR): checklist and explanation. Annals of internal medicine, 169(7), 467-473].

For the less informed reader it is not so clear what have been done and the abstract is not clear.

Do authors intend to continue and perform a systematic review?

Did authors registred the review protocol? If not publicly available, may authors provide details about how to access it?

We understand that in a scoping review it is not mandatory a critical appraisal or risk of bias assessment of the included studies. Nevertheless, in a qualitaty appraisal like this, a paragraph stating the overall quality might be interesting to future readers and putative citations. Moreover, when the manuscript intends to be used for decision-making or to inform future research directions, we would like to see described the bias assessment (it means, whether the included studies should be believed, being different from assessing the methodological quality).

Reviewer 3 Report

Overall the contents of the manuscript are good and can be a reference for further reviews in developing countries to be used as inclusion variables

Author Response

Thank you for your comment. We appreciate your feedback.

Reviewer 4 Report

This manuscript is excellent and is an essential addition to the literature. I enjoyed reading and critically evaluating this manuscript. However, a few points are needed to consider in this article,

Abstract: This part is perfect regarding the study's topic, result, and conclusion. Keywords: simple for readers to know the content of the study.

Introduction: The introduction provides a good, generalized background of the topic that quickly gives the reader an appreciation of the impact of dental care programs on individuals and their families: A scoping review. However, to make the introduction more substantiated, the authors may add more references related to the need of the study. Overall this part is well-explained.  

Material and method: This study has considered every important point required. Techniques such as PICO and inclusion-exclusion criteria were clearly and well explained. 

Results: Well-written results and tables are self-explanatory. Data are well presented, and no need for any supplementary figures or tables.

Discussion: This part of the study is well explained along with the result of the study.   

The conclusion is short but relevant to the study.

References are appropriately marked, and no duplication is seen.

Author Response

Thank you for your comments. We added more references to the introduction.